# Analysis of Deformation Behavior and Microstructure Changes for α/β Titanium Alloy at Elevated Temperature

**Wenhua Yang [1,*], Wei Ji [1], Zhaohui Zhou [1], Aiguo Hao [1], Linxin Qing [1], Hualei Hao [1] and Jianwei Xu [2,*]**

[1] Institute of Machinery Manufacturing Technology, China Academy of Engineering Physics, Mianyang 621900, China; xjw9310@126.com (W.J.); gaoyanssd@163.com (Z.Z.); zwyhag@163.com (A.H.); 13262778535@163.com (L.Q.); lifepresenting@163.com (H.H.)

[2] School of Materials Science and Engineering, Northwestern Polytechnical University, Xi'an 710072, China

[*] Correspondence: ywh060962@126.com (W.Y.); jianwei_xu@nwpu.edu.cn (J.X.); Tel.: +86-816-2485668 (W.Y.)

**Abstract:** In this paper, the isothermal compressive behavior of Ti-6.5Al-3.5Mo-1.5Zr-0.3Si titanium alloy was investigated on a Gleeble-3500 simulator in the temperature range from 1073 to 1373 K at an interval of 50 K (while the phase transus temperature is approximately 1273 K) and the strain rate range of 0.001–10 s$^{-1}$. Microstructure evolution and deformation behavior were investigated. The typical flow softening behavior during deformation is observed, which can be explained by the deformation heating effect and microstructure changes. The deformation heating effect is influenced by strain rate and deformation temperature, and it increases with the increasing strain rate and decreasing deformation temperature. In the α + β phase field, the fractions of the primary α phase decrease with the increase of deformation temperature and strain rate. In this case, dynamic recovery may be the main mechanism for microstructure evolution based on the electron back-scatter diffraction (EBSD) analysis. The fully phase transformation occurs above the β transus temperature, which is governed by Burgers orientation relations. The Zener–Hollomon parameter with an exponent-type equation was used to intuitively describe the effects of the deformation temperatures and strain rates on the flow stress behaviors. Furthermore, the influence of strain was incorporated in the constitutive analysis. A fourth-order polynomial was ideally matched to represent the influence of strain. In consequence, the constitutive equation of Ti-6.5Al-3.5Mo-1.5Zr-0.3Si titanium alloy including the phase transus and compensation of the strain was developed based on the experimental results throughout the deformation process. The results indicated that the correlation coefficient (R), root mean square error (RMSE), and the average absolute relative error (AARE) were calculated to be 0.987, 3.585 MPa, and 9.62% in the single-phase region and 0.979, 18.78 MPa, and 9.16% in the duplex-phase region, respectively. Hence, the constitutive model proposed in this research can provide accurate and precise theoretical prediction for the flow stress behavior of Ti-6.5Al-3.5Mo-1.5Zr-0.3Si titanium alloy.

**Keywords:** Ti-6.5Al-3.5Mo-1.5Zr-0.3Si alloy; flow behavior; microstructure evolution; constitutive modeling



## 1. Introduction

The application of titanium alloy, with various metal elements that are able to significantly strengthen its mechanical and physical properties, has been gradually increased in aerospace, energy, and chemical industries due to their combination of excellent strength to weight ratio and superior crack and fatigue propagation resistance [1,2]. However, with the perpetual improvement of the engine thrust and the increase of compressor pressure ratio, the materials are required to fit more intricate working conditions. More rigorous demands of titanium alloy are taking into consideration so as to meet further application. Ti-6.5Al-3.5Mo-1.5Zr-0.3Si alloy, based on the Ti-6Al matrix beta isomorphous element Mo, neutral element Zr, and beta eutectoid elements Si that are doped, has higher strength and

more splendid thermal stability, particularly at elevated temperatures in comparison with universal Ti-6Al-4V and other titanium alloys [3–6].

Currently, comprehensive manufacture techniques for the titanium alloy, such as forging, punching, and cutting, have been introduced to prepare military aircraft, high-performance aero-engines, military armor, and so forth. It is extensively accepted that mechanical properties of alloys are quite susceptible to the microscopic structure, whereas the microstructure of materials is significantly dependent on the hot deformation parameters [7,8]. As a consequence, the flow stress behavior and microstructural evolution on Ti-6.5Al-3.5Mo-1.5Zr-0.3Si titanium alloy should be investigated under variable deformation parameters, and the optimum temperatures and interrelated parameters during hot deformation are ascertained to meet the standard demands of products.

Noticeable efforts have been made to investigate the flow stress behavior and microstructural evolution of titanium alloys. For example, Peng et al. [9] studied the effect of strain rate, temperature, and strain on the flow behavior for TC4-DT alloy by establishing a constitutive equation with a view to the variation of the material constants affected by strain. In addition, the relationship between flow stress behavior and phase transformation was mentioned in their work. Qu et al. [10] illustrated the effect of deformation parameters for Ti-5Al-5Mo-5V-1Cr-1Fe alloy on the mechanical properties. The material constants of the constitutive relationship model were determined by using the data gathered from macroscopic uniaxial loading responses during hot deformation. The results demonstrated that the processing parameters have a significant effect on the softening mechanism. Zhou et al. [11] researched the characterization of hot workability for Ti80 alloy by integrating processing maps and establishing a constitutive relationship. Xu et al. [12] studied the flow softening and microstructure evolution of Ti-17 alloy and found that the microstructure evolution of alpha and beta phases together influence flow softening. Moreover, the globularization of the alpha phase and recrystallization of the beta phase result in flow softening. A dynamic softening map has also been established based on the dislocation density evolution, and the result shows that a dynamic recrystallization (DRX) process tends to occur at low strain rates.

The above-mentioned research work on duplex phase titanium alloys are worth mentioning for recognizing the flow stress behavior, microstructure, and texture evolution of the material during hot deformation [10–13]. The relevant studies can provide reference for the current work. The main target of the present work is to systematically evaluate the deformation behavior, microstructure changes, and constitutive behavior under a wide range of deformation temperature and strain rate with the consideration of phase transformation. The microstructure evolution of Ti-6.5Al-3.5Mo-1.5Zr-0.3Si alloy was studied by employing the light microscope (LM), scanning electron microscope (SEM), and electron back-scatter diffraction (EBSD). The role of the deformation parameters factors such as the deformation temperature (T), strain rate ($\dot{\varepsilon}$), true strain ($\varepsilon$), and deformation activation energy (Q) on the flow stress behavior are analyzed, respectively. In addition, the constitutive model with high prediction accuracy will be established.

## 2. Experimental Materials and Procedure

The raw material used in this study is Ti-6.5Al-3.5Mo-1.5Zr-0.3Si titanium alloy bar with the chemical composition listed in Table 1. The original microstructure of the alloy is shown in Figure 1. It can be observed from this figure that this alloy has a typical bi-modal microstructure, which consists of a combination of lamellar dispersed in equiaxed grains matrix homogeneously. The β transus temperature measured by metallographic analysis was around 1273 K. The isothermal compression tests were conducted on a Gleebe-3500 hot-simulator (Data Sciences International Inc., DE, USA) in the deformation temperature range of 1073–1373 K and the strain rates range of 0.001–10 s$^{-1}$. The height reduction of the specimen was 60%. The graphite lubricant was applied to minimize the deformed friction effects during the deformation process. Cylindrical specimens machined from bars, with a diameter of 8 mm and a height of 12 mm, were resistance heated by thermocouples

feedback. The specimens were heated to deformation temperature at a heating rate of 5 K/s and kept for 3 min to homogenize the temperature in the sample. Graphite powder was applied between specimens and anvils in order to reduce die friction. Temperature was monitored by a thermocouple that was welded in the center of surface of the specimen during deformation. The deformation data were automatically recorded by a thermo-simulator system. After hot compression, in order to preserve the hot-deformed structures, the specimens were quenched into water immediately. The deformed specimens were sectioned parallel to the compression axis for microstructure analysis. The samples for metallographic examination were prepared by mechanically polishing and etching with a solution consisting of 5 vol.% HF, 15 vol.% HNO$_3$, and 80 vol.% H$_2$O. The center of the specimens was chosen as the observation point to observe microstructure changes. The observations of microstructural evolution were carried out by the LM (Olympus PM-T3, Olympus Corporation, Tokyo, Japan), SEM (TESCAN VEGA 3 LMU, TESCAN ORSAY HOLDING, Brno, Czech Republic), and EBSD (ZEISS Gemini 500 with Nordlys Nano Detector, Carl Zeiss AG, Jena, Germany). In the EBSD tests, step size was set as 0.5 μm, and the EBSD data were analyzed by Channel 5 software (OXFORD INSTRUMENTS, Oxford, UK).

**Table 1.** Chemical composition of the Ti-6.5Al-3.5Mo-1.5Zr-0.3Si alloy (wt %).

| Al | Mo | Zr | Si | Fe | Ti |
|------|------|------|------|-------|----------|
| 6.48 | 3.34 | 1.74 | 0.24 | 0.067 | balanced |

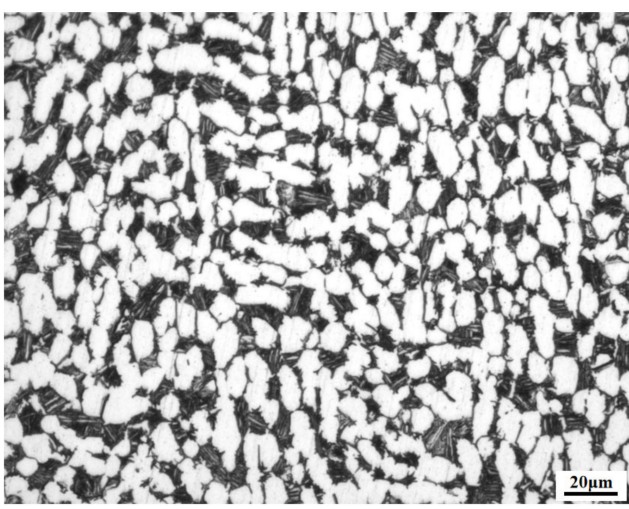

**Figure 1.** The initial microstructure of as-received material.

### 3. Results and Discussion

*3.1. Deformation Behavior and Microstructure Evolution*

3.1.1. Flow Behavior

The true stress–strain curves based on the data collected from hot compressed tests of Ti-6.5Al-3.5Mo-1.5Zr-0.3Si alloy at different deformation temperatures and strain rates are shown in Figure 2. Obviously, the figures show that the flow stress curves are split into three stages under fixed temperature and strain rate with the increasing of the strain. To be specific, at the beginning of hot compressing, the flow stress increases sharply with the increasing of strain due to the sake of the foremost role of work hardening. Along with the increase of strain, the growth rate of true stress slows down attributing to dynamic softening, and then, the flow stress curve reaches the peak value. The softening results in the obvious decrease of the flow stress after peak stress, which could be obtained from most flow curves. With the accumulation of strain during hot deformation, the flow stress then keeps a steady state as a result of the balance of dynamic softening and

work-hardening effects. Discontinuous yielding is observed at 1323 K. Such behavior is associated with the increasing mobile dislocation at grain boundary, and it is strengthened as the increasing temperature and strain rate [14,15]. It can also be seen from Figure 2 that the steady state of the flow stress of Ti-6.5Al-3.5Mo-1.5Zr-0.3Si titanium alloy was apparently affected by deformation temperature and strain rate during hot processing. At lower deformation temperatures, the flow stress curves exhibit dramatic work hardening followed by significant flow softening; at last, a steady state is reached. In contrast, the flow stress curves present conspicuous steady-state characteristics during higher deformation temperatures. As shown in Figure 2, the flow stress curves exhibit a typical flow behavior with the softening, including a single peak followed by a steady-state flow upward β transus temperature. The steady-state stress ($\sigma_{ss}$) is 37 MPa at 1223 K/0.01 s$^{-1}$, and the value changes to 185 MPa when the strain rate increases to 10 s$^{-1}$. Hence, it can be considered that the steady flow stress is extremely sensitive to strain rate and temperature, or the occurrence of steady state on the flow stress behavior would be promoted with the increase of deformation temperature, while the value of steady-state stress decreases with the reduction of strain rate.

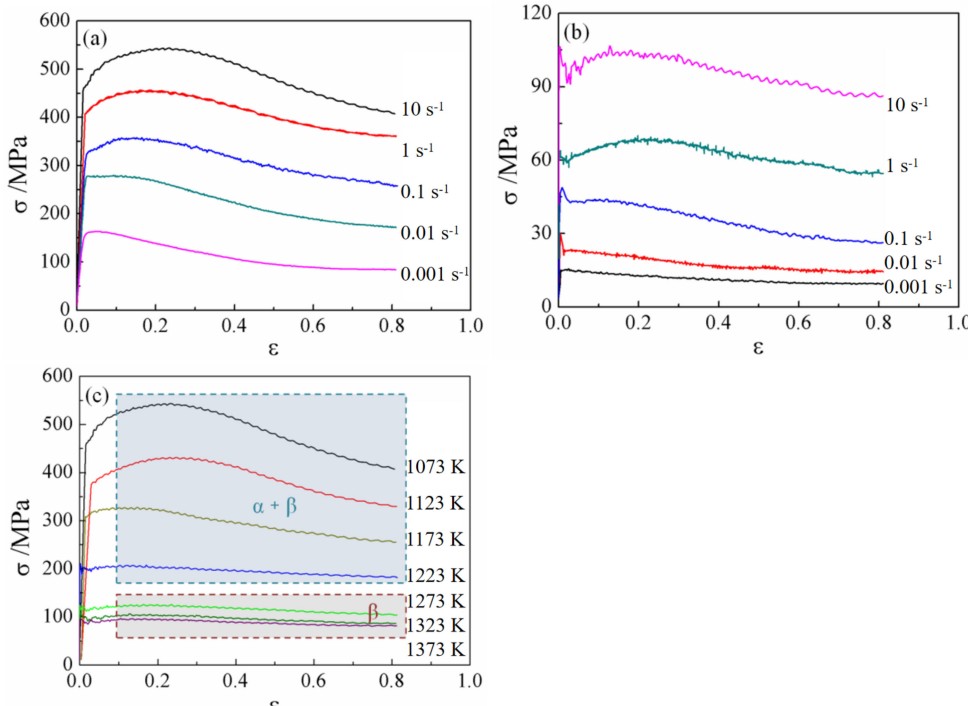

**Figure 2.** True stress-strain curves of Ti-6.5Al-3.5Mo-1.5Zr-0.3Si titanium ally with various strain rates and deformation temperatures: (**a**) 1073 K, (**b**) 1323 K, and (**c**) 10 s$^{-1}$.

The softening stresses ($\Delta\sigma = \sigma_p - \sigma_{ss}$) at different deformation temperatures and strain rates, which indicate the index of dynamic softening effect, are presented in the form of a three-dimensional image, as shown in Figure 3. Apparently, the softening stress increases significantly with the decreasing of temperature or the increasing of strain rate. While at a low strain rate, the softening stress is kept almost constant during different deformation temperatures. In other words, the value of softening stress is sensitive to deformation temperature and strain rate. For titanium alloys, deformation heat and microstructure changes are major factors to cause flow softening [16,17].

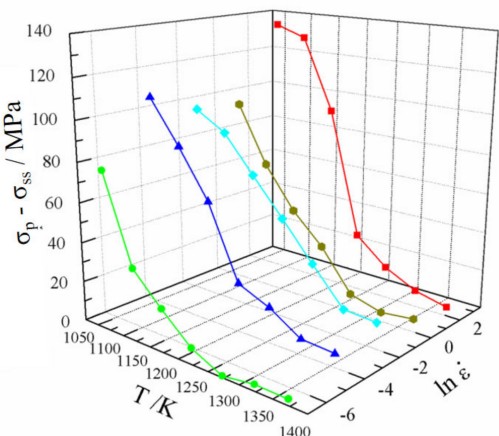

**Figure 3.** The relationship between strain rate, temperature, and flow softening.

### 3.1.2. Deformation Heating

The deformation heating effect can lead to temperature rise during hot compression, which will reduce the deformation resistance of titanium alloy. Thus, the deformation heating effect is the main reason for the flow-softening behavior. The deformation heating effect is influenced by deformation temperature and strain rate, and temperature rise can be calculated by the following equation [18]:

$$T_R = \frac{\lambda \int_0^\varepsilon \sigma d\varepsilon}{\rho \cdot c} \tag{1}$$

in which $T_R$ is temperature rise, $\lambda$ is transformation coefficient, $\varepsilon$, $\sigma$, $c$, $\rho$, and $\lambda$ are strain, flow stress, specific heat capacity, density, and transformation coefficient, respectively. The value range of $\lambda$ is as follows [18]:

$$\lambda = \begin{cases} 0 & \dot{\varepsilon} = 0.001 \\ 0.33 & \dot{\varepsilon} = 0.01 \\ 0.66 & \dot{\varepsilon} = 0.1 \\ 0.88 & \dot{\varepsilon} = 1 \\ 0.98 & \dot{\varepsilon} = 10 \end{cases}. \tag{2}$$

The transformation coefficient $\lambda$ is a constant and is determined by strain rate. The hot compression is considered as an isothermal process when the strain rate is 0.001 s$^{-1}$, and thus $\lambda = 0$. The $\lambda$ increases with the increasing strain rate. The values $\lambda$ are 0.33, 0.66, 0.88, and 0.98 under strain rates of 0.01, 0.1, 1, and 10 s$^{-1}$, respectively. Temperature rises are calculated, and the results are listed in Table 2. From the calculated results, temperature rise increases significantly at lower deformation temperature and higher strain rate. For example, the largest temperature rise is 75.58 K at a deformation temperature of 1073 K and strain rate of 10 s$^{-1}$. In contrast, the temperature rise is only 1.98 K at a deformation temperature of 1373 K and a strain rate of 0.01 s$^{-1}$. Flow softening is more obvious under lower deformation temperature and higher strain rate.

**Table 2.** Deformation heating effect—the temperature rise (K) of the material under different deformation conditions.

| Deformation Temperature (K) | Strain Rate (s$^{-1}$) | | | | |
|---|---|---|---|---|---|
| | 0.001 | 0.01 | 0.1 | 1 | 10 |
| 1073 | 0 | 36.54 | 42.15 | 56.23 | 75.58 |
| 1123 | 0 | 29.05 | 35.36 | 45.25 | 60.46 |
| 1173 | 0 | 22.35 | 28.11 | 33.32 | 43.56 |
| 1223 | 0 | 13.21 | 16.01 | 24.23 | 31.21 |
| 1273 | 0 | 6.38 | 8.97 | 15.21 | 18.33 |
| 1323 | 0 | 2.12 | 4.07 | 7.25 | 10.21 |
| 1373 | 0 | 1.98 | 3.56 | 7.01 | 8.65 |

### 3.1.3. Microstructure Evolution

Microstructure morphology presents different characteristics under various deformation conditions, and its changes can explain deformation behavior, such as flow softening. Microstructure changes cause a remarkable influence on the mechanical properties of duplex titanium alloys [19]. Figure 4 shows the microstructure morphology of the Ti-6.5Al-3.5Mo-1.5Zr-0.3Si alloy deformed under different conditions. In order to observe carefully microstructure evolution, the SEM pictures are shown in Figure 5, which will provide larger magnification and clearer microstructure morphology.

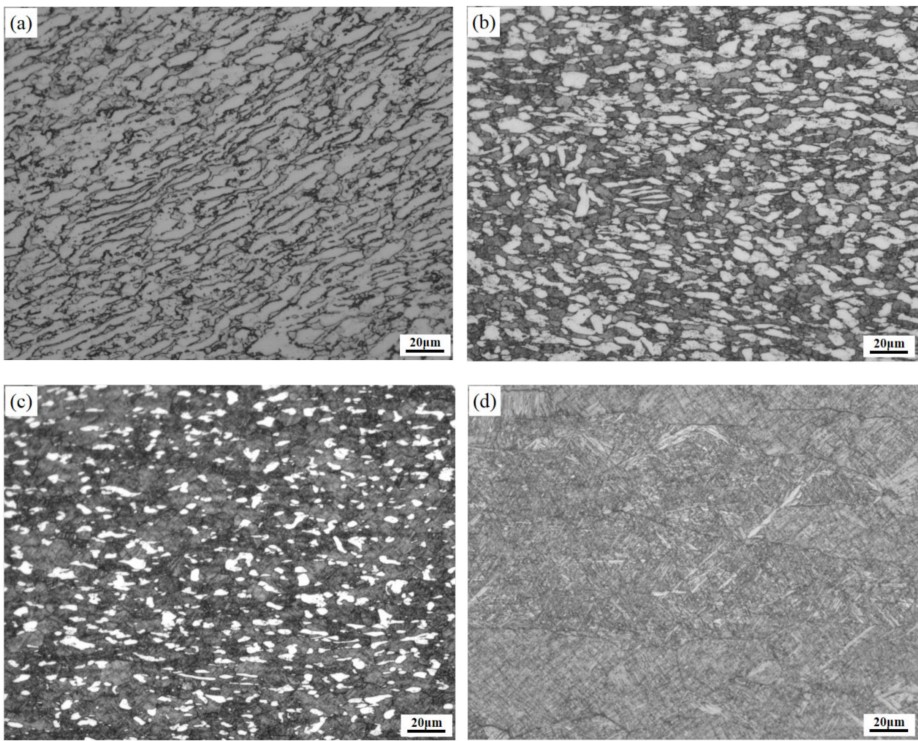

**Figure 4.** The light microscope (LM) microstructure of Ti-6.5Al-3.5Mo-1.5Zr-0.3Si alloy after deformation at: (**a**) 1123 K/0.1 s$^{-1}$, (**b**) 1223 K/0.1 s$^{-1}$, (**c**) 1223 K/10 s$^{-1}$, and (**d**) 1323 K/0.1 s$^{-1}$.

It can be seen from Figures 4a–c and 5a–c that the deformed microstructure consist of primary α phase and transformed β matrix when the samples are deformed below β transus temperature. In this case, microstructure evolution is insensitive to deformation temperature and strain rate. The fractions of the primary α phase decrease with the increase of deformation temperature at the same strain rate. For instance, the content of the alpha phase at 1123 K is significantly higher than that at 1223 K. The quantitative analysis shows that the volume fraction of the alpha phase at 1123 K is over 85%, while the scale is only

45% at 1223 K. The volume fractions of the primary α phase are also different under various strain rates, and it can be verified by the comparison between Figure 4b,c and Figure 5b,c. Such phenomenon is caused by the deformation heating effect. From Table 2, the difference of temperature rise is 15.2 K under strain rates of 0.1 s⁻¹ and 10 s⁻¹, which results in a different content of primary α phase. As shown in Figures 4d and 5d, a full phase transformation is observed when the deformation is conducted above the β phase transus temperature. In this case, a fully transformed β matrix is obtained.

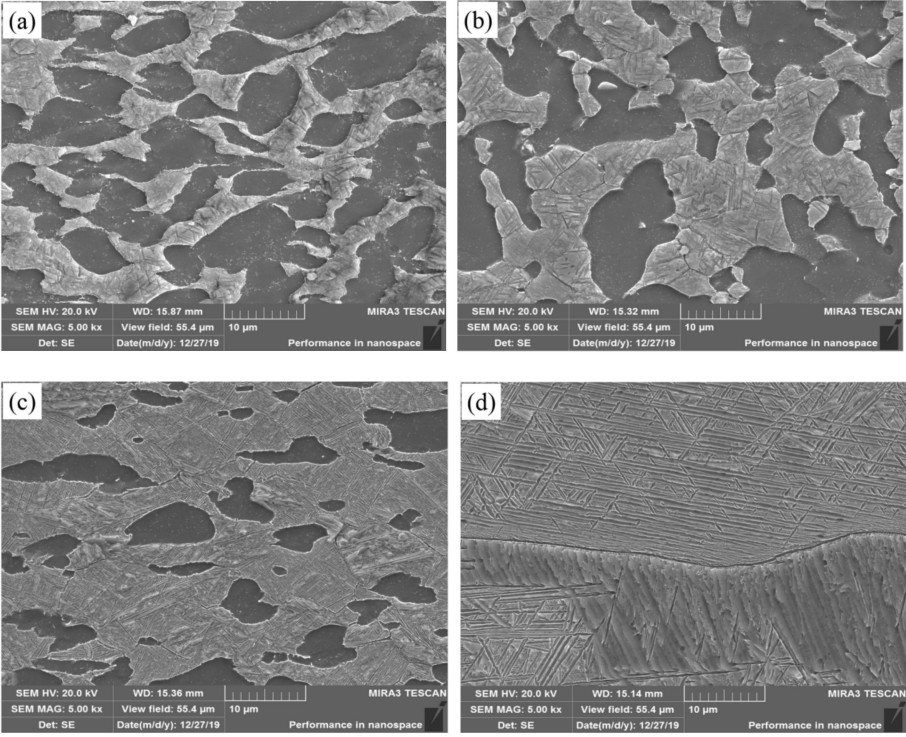

**Figure 5.** The SEM microstructure of Ti-6.5Al-3.5Mo-1.5Zr-0.3Si alloy after deformation at (**a**) 1123 K/0.1 s⁻¹, (**b**) 1223 K/0.1 s⁻¹, (**c**) 1223 K/10 s⁻¹, and (**d**) 1323 K/0.1 s⁻¹.

Figure 6 illustrates the inverse pole figure (IPF) maps for material deformed at 1123, 1223, and 1323 K under a strain rate of 0.1 s⁻¹. The change of crystal orientation of the alpha phase is observed from Figure 6. At 1123 K, a mass of primary alpha phase is reserved. The uneven color within an alpha grain is observed, which indicates that intracrystalline orientation has changed. There are the low-angle boundaries in the alpha phase, which can be verified by the distribution of boundary misorientation, as shown in Figure 7a. In this case, dynamic recovery may be the main mechanism for microstructure evolution. Part of the primary alpha phase is reserved, and more secondary alpha phase is separated out when the deformation temperature increases to 1223 K. For the primary alpha phase, it has a similar process of microstructure evolution to that at 1123 K. A typical characteristic of phase transformation is found for the secondary alpha phase. Except for low-angle boundary misorientation, the boundary misorientation nearby 60° and 90° appear, as shown in Figure 7b. This phenomenon is more obvious at 1323 K, which can be observed in Figure 7c. Such changes can be explained by variation selection, in which alpha and beta phases are governed by Burgers orientation relationships: $\{0001\}_{\alpha}//\{110\}_{\beta}$, $<11–20>_{\alpha}//<111>_{\beta}$ when the alpha phase is separated out from the beta matrix [20].

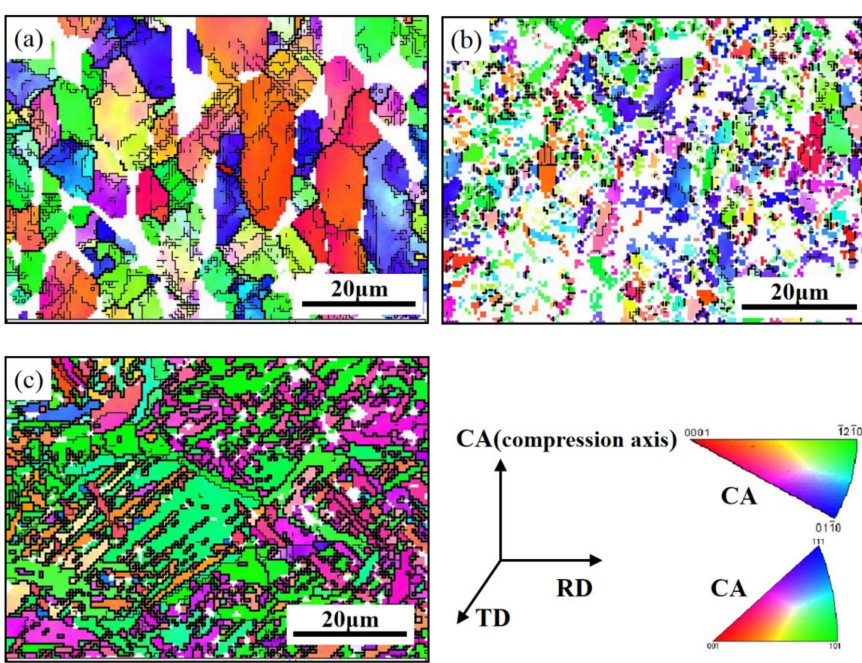

**Figure 6.** The inverse pole figure (IPF) maps of Ti-6.5Al-3.5Mo-1.5Zr-0.3Si alloy after deformation at:
(**a**) 1123 K/0.1 s$^{-1}$, (**b**) 1223 K/0.1 s$^{-1}$, and (**c**) 1323 K/0.1 s$^{-1}$. The bold black lines correspond to
high-angle boundaries with misorientation over 15°, while the thin black lines represent low-angle
boundaries with misorientation between 2° and 15°.

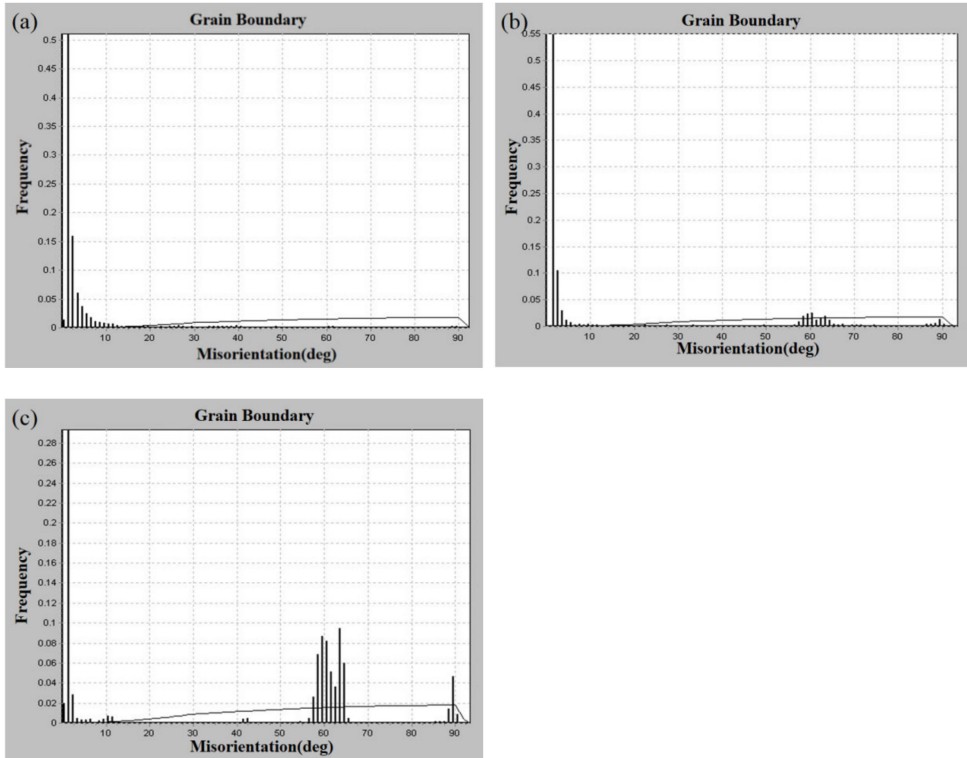

**Figure 7.** The distribution of boundary misorientation of Ti-6.5Al-3.5Mo-1.5Zr-0.3Si alloy after deformation at: (**a**)
1123 K/0.1 s$^{-1}$, (**b**) 1223 K/0.1 s$^{-1}$, and (**c**) 1323 K/0.1 s$^{-1}$.

Figure 8 shows the Schmid factor maps for Ti-6.5Al-3.5Mo-1.5Zr-0.3Si alloy under
different deformation conditions. The maps exhibit the Schmid factor distributions of
basal slip {0001} <11–20>, prismatic slip {1–100} <11–20>, and pyramidal slip {1–101} <11–

20>. Under three deformation conditions, the Schmid factor of pyramidal slip is largest, prismatic slip comes second, and basal slip is smallest. The ease or complexity of activation of the slip systems are confirmed according to the Schmid factor: pyramidal slip > prismatic slip > basal slip. Pyramidal slip is easier to activate in contrast to prismatic slip and basal slip. Such phenomenon is related to the structural features of the alpha phase in titanium alloy, where the axial ratio (c/a) is about 1.587. The prismatic or pyramidal slips are the easier to activate when c/a < 1.633 [21]. The information about the Schmid factor is helpful to assist readers to understand microstructure changes during deformation.

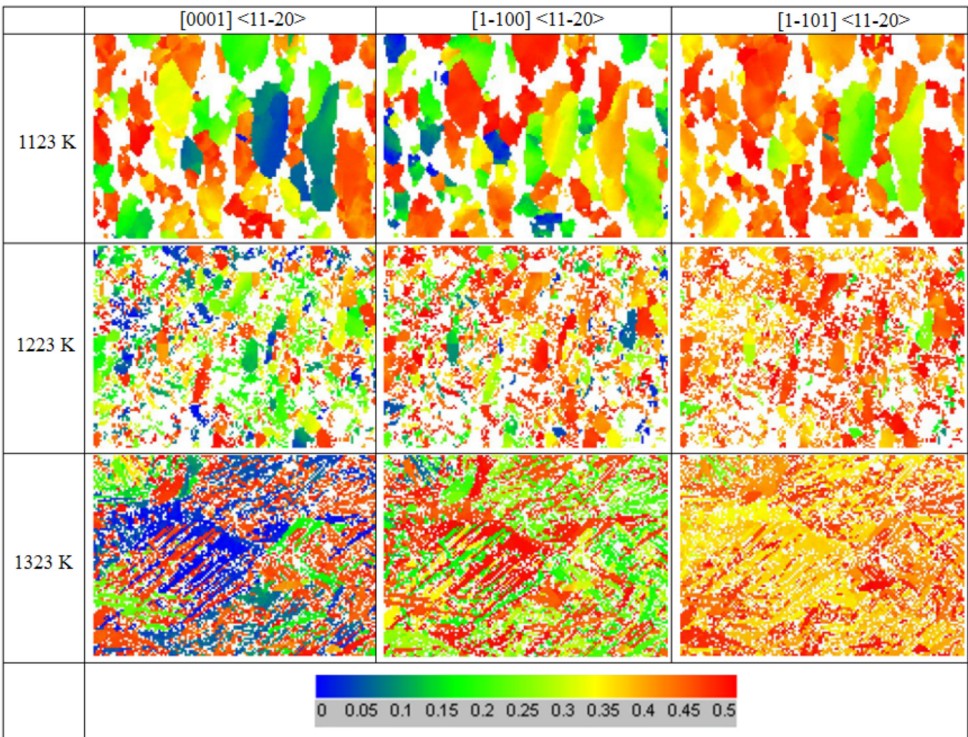

**Figure 8.** The Schmidt factors of Ti-6.5Al-3.5Mo-1.5Zr-0.3Si alloy after deformation under various conditions.

### 3.2. Constitutive Relation

### 3.2.1. Constitutive Model Development

Generally, the primary target of the established constitutive model is to describe the plastic deformation behavior during hot deformation and can predict the flow stress in the numerical simulation process. At present, the constitutive models proposed by the peer researchers can generally be divided into three categories: artificial neural network models, physical-based models, and phenomenological models [22,23]. As the high deformation temperature of alloys is controlled by thermal activation, the temperature and strain rate dependence of flow stress is generally expressed in terms of extensively utilized Arrhenius kinetic equation, containing the impact of thermal activation. In this model, the effects of strain rate, deformation temperature, and strain on the flow stress behavior of the present alloy at elevated temperatures are taken into consideration. Based on the true stress–true strain data gathered from the hot compression test, the constitutive relationship equation for Ti-6.5Al-3.5Mo-1.5Zr-0.3Si alloy is established in terms of the employment of the Arrhenius-type equation. The dependence of the flow stress on the deformation temperature and strain rate can be designated as [22]:

$$\dot{\varepsilon} = AG(\sigma)e^{-Q/RT} \tag{3}$$

where:

$$G(\sigma) = \begin{cases} \sigma^{n'} & \alpha\sigma < 0.8 \\ \exp(\beta\sigma) & \alpha\sigma > 1.2 \\ [\sinh(\alpha\sigma)]^n & \text{for all } \sigma \end{cases} \tag{4}$$

in which $\sigma$ is true stress (MPa), $\dot{\varepsilon}$ is strain rate (s$^{-1}$), A, $\alpha$, n', n, and $\beta$ are material constants, Q is the deformation activation energy (kJ/mol), R is the universal gas constant (8.314 J/mol·K), and T is the absolute temperature (K). The stress multiplier $\alpha$ is defined as $\alpha = \beta/n'$. Meanwhile, the commixture of strain rate and temperature effects on the deformation behavior of materials can be expressed by Zener–Hollomon parameter (Z) with an exponent-type equation [24], which is given as $z = \dot{\varepsilon}e^{Q/RT}$ in Equation (3) and is used to describe the flow stress behavior. Sellars and McTegart [25] indicated that $G(\sigma) = [\sinh(\alpha\sigma)]^n$ is able to accurately describe hot plastic deformation behavior and applies to both low stress and high stress conditions. Thus, the Arrhenius equation is represented as:

$$z = \dot{\varepsilon}e^{Q/RT} = A[\sinh(\alpha\sigma)]^n \tag{5}$$

In this research, the influence of strain on different material constants in the constitutive model, including $\beta$ transus factors, is studied. Based on the experimental results acquired from the hot compression tests under various conditions, the true strain of 0.6 was firstly taken as an example to demonstrate the calculative procedures of material constants. Subsequently, the material constants employed in the constitutive model can be determined according to the regression analysis of the experimental data.

For the low stress level ($\alpha\sigma < 0.8$) and high stress level ($\alpha\sigma > 1.2$), the values of $G(\sigma)$ are substituted into Equation (3) and the relationships can be obtained, as shown in Equations (6) and (7), respectively.

$$\dot{\varepsilon} = A_1 \sigma^{n'} e^{Q/RT} \ (\text{for } \alpha\sigma < 0.8) \tag{6}$$

$$\dot{\varepsilon} = A_2 e^{\beta\sigma} e^{Q/RT} (\text{for } \alpha\sigma > 1.2) \tag{7}$$

where $A_1$ and $A_2$ are material constants, which are independent of the deformation temperatures. In Equation (5), the constitutive model of Ti-6.5Al-3.5Mo-1.5Zr-0.3Si titanium alloy under various deformation conditions can be established, if the parameters, such as A, n, $\alpha$ and Q, are acquired. By taking the natural logarithm of Equations (5)–(7) on both sides, the following equations can be obtained:

$$\ln\dot{\varepsilon} = \ln A_1 + n'\ln\sigma - \frac{Q}{RT} \tag{8}$$

$$\ln\dot{\varepsilon} = \ln A_2 + \beta\sigma - \frac{Q}{RT} \tag{9}$$

$$\ln Z = \ln\dot{\varepsilon} + \frac{Q}{RT} = \ln A + n\ln[\sinh(\alpha\sigma)]. \tag{10}$$

According to Equations (8) and (9), $n' = \partial\ln\dot{\varepsilon}/\partial\ln\sigma$ and $\beta = \partial\ln\dot{\varepsilon}/\partial\sigma$. The values of $\beta$ and $n'$ can be obtained by means of regression analysis based on the flow stress data at different deformation temperatures and strain rates and a certain strain (e.g., $\varepsilon = 0.6$). Based on the ln$\dot{\varepsilon}$-$\sigma$ and ln$\dot{\varepsilon}$-ln$\sigma$ curves as shown in Figures 9 and 10, the slopes of the linear fitting values will be $\beta$ and $n'$, respectively. The majority of titanium alloys exhibit different flow stress behaviors in diverse phase regions, and such behavior significantly affects the values of material constants. Hence, the material constant of $\alpha$ is separately calculated to be 0.009 for the $\alpha + \beta$ phase region and 0.025 for the $\beta$ phase region.

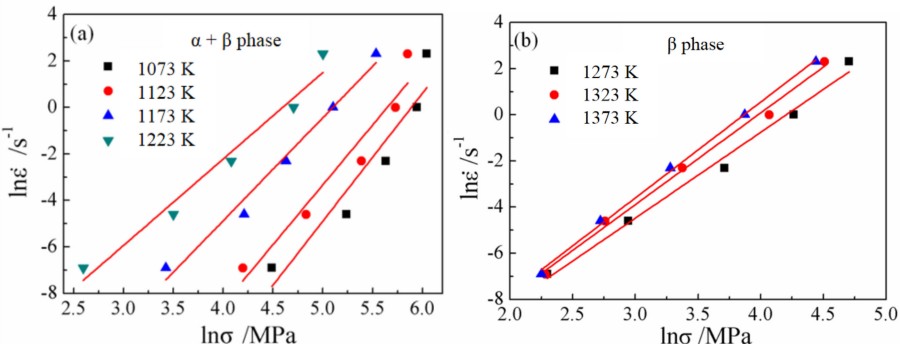

**Figure 9.** Relationships between lnε̇ and lnσ at a strain of 0.6 in different phase region. (**a**) α + β phase region and (**b**) β phase region.

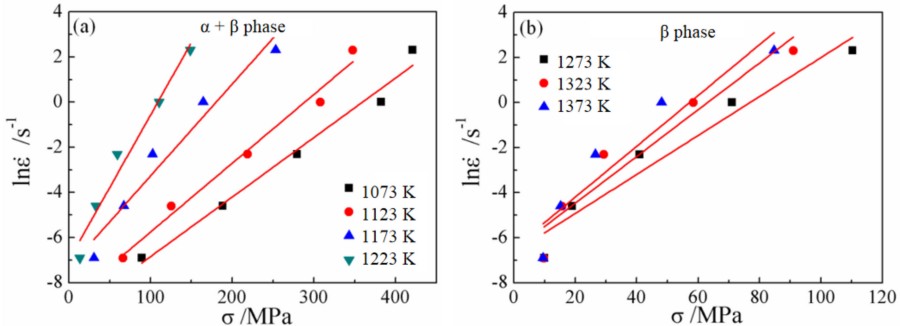

**Figure 10.** Relationships between lnε̇ and σ at a strain of 0.6 in different phase region: (**a**) α + β phase region and (**b**) β phase region.

On the basis of the obtained α constant, the relation curves of $\ln[\sinh(\alpha\sigma)]$ vs. $\ln\dot{\varepsilon}$ and $\ln[\sinh(\alpha\sigma)]$ vs. $1000/T$ are shown in Figures 11 and 12, respectively. The deformation activation energy (Q) can be calculated according to $Q = R\frac{\partial(\ln(\sinh(\alpha\sigma)))}{\partial(1/T)} / \frac{\partial(\ln(\sinh(\alpha\sigma)))}{\partial(\ln\dot{\varepsilon})}$, which is deduced from Equation (10). The calculated values of Q are 507.1 kJ/mol for the α + β phase region and 173.027 kJ/mol for the β phase region, respectively. Therefore, the constitutive relationship of Ti-6.5Al-3.5Mo -1.5Zr-0.3Si titanium alloy with various hot working conditions across a β transus temperature can be presented as:

$$Z = \dot{\varepsilon}e^{507100/8.314T} = 2.13 \times 10^{21}[\sinh(0.00835\sigma_m)]^{3.057} \tag{11}$$

$$Z = \dot{\varepsilon}e^{173027/8.314T} = 2.32 \times 10^6[\sinh(0.024\sigma_m)]^{2.935}. \tag{12}$$

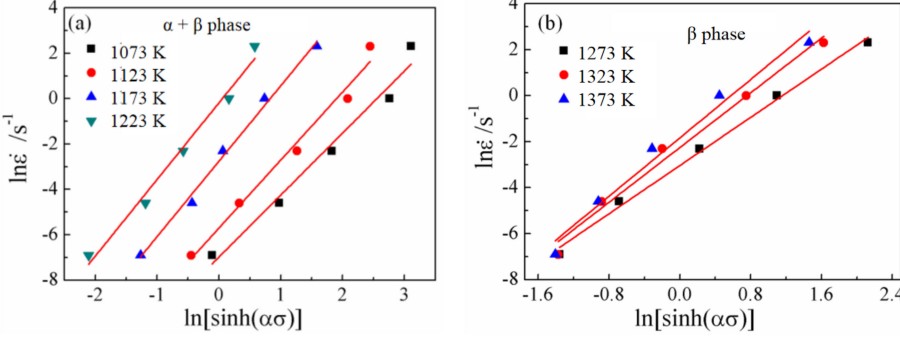

**Figure 11.** Relationships between ln[sinh(ασ)] and lnε̇ at (**a**) the α + β phase region and (**b**) the β phase region.

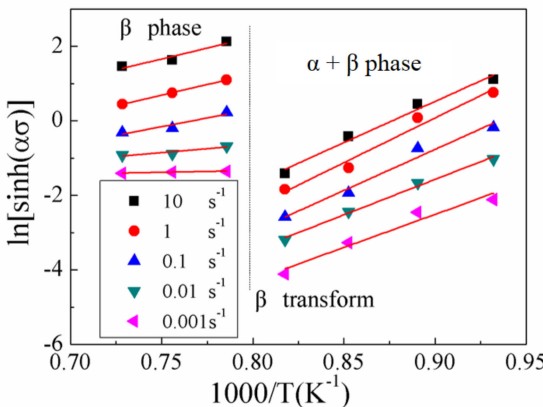

**Figure 12.** Relationships between $1000/T$ and $\ln[\sinh(\alpha\sigma)]$.

### 3.2.2. Effect of Strain on Material Constants

It is considered that the effect of strain on flow stress under high temperature is of no importance and thus would usually be ignored in Equation (5) [26]. However, the effects of the strain on the material constants (i.e., $\alpha$, n, Q, and $ln$A) are significant among the entire strain range, as shown in Figure 13. Therefore, the compensation of strain should also be taken into consideration so as to enhance the accuracy of the established constitutive model. The effect of strain was recommended by assuming that each material constant was polynomial functions of strain [26]. In this work, the values of material constants in the constitutive equation are calculated out in various true strain ranging from 0.1 to 0.65 at the interval of 0.05. In order to improve the relevance and accuracy of the fitting curves, a fourth-order polynomial is utilized to demonstrate the effect of strain on various material constants as given by Equation (13). The values of polynomial fitting coefficients of $\alpha$, n, Q, and $ln$A in the $\alpha + \beta$ and $\beta$ phase regions are provided in Tables 3 and 4, respectively.

$$
\begin{aligned}
\alpha &= D_0 + D_1\varepsilon + D_2\varepsilon^2 + D_3\varepsilon^3 + D_4\varepsilon^4 \\
n &= E_0 + E_1\varepsilon + E_2\varepsilon^2 + E_3\varepsilon^3 + E_4\varepsilon^4 \\
Q &= F_0 + F_1\varepsilon + F_2\varepsilon^2 + F_3\varepsilon^3 + F_4\varepsilon^4 \\
\ln A &= G_0 + G_1\varepsilon + G_2\varepsilon^2 + G_3\varepsilon^3 + G_4\varepsilon^4
\end{aligned}
\tag{13}
$$

**Table 3.** Parameters of $\alpha$, n, Q, and $ln$A at different strains in the $\alpha + \beta$ phase region.

| $\alpha$ | $n$ | $Q$ | $ln$A |
|---|---|---|---|
| $D_0 = 0.04612$ | $E_0 = 4.29608$ | $F_0 = 611.52676$ | $G_0 = 60.08459$ |
| $D_1 = -0.08725$ | $E_1 = -6.7472$ | $F_1 = -177.11583$ | $G_1 = -21.13108$ |
| $D_2 = 0.25505$ | $E_2 = 12.63285$ | $F_2 = -402.57064$ | $G_2 = -34.79325$ |
| $D_3 = -0.28671$ | $E_3 = -9.74902$ | $F_3 = 1033.56546$ | $G_3 = 102.12099$ |
| $D_4 = 0.11812$ | $E_4 = 2.75381$ | $F_4 = -601.28837$ | $G_4 = -61.53978$ |

**Table 4.** Parameters of $\alpha$, n, Q, and $ln$A at different strains in the $\beta$ phase region.

| $\alpha$ | $n$ | $Q$ | $ln$A |
|---|---|---|---|
| $D_0 = 0.10981$ | $E_0 = 3.60527$ | $F_0 = 222.5184$ | $G_0 = 19.02768$ |
| $D_1 = -0.19703$ | $E_1 = -4.44552$ | $F_1 = -372.822$ | $G_1 = -36.5263$ |
| $D_2 = 0.74282$ | $E_2 = 12.00194$ | $F_2 = 1993.765$ | $G_2 = 197.2343$ |
| $D_3 = -1.05826$ | $E_3 = -15.8513$ | $F_3 = -3674.72$ | $G_3 = -360.061$ |
| $D_4 = 0.50748$ | $E_4 = 8.45562$ | $F_4 = 1957.719$ | $G_4 = 190.2746$ |

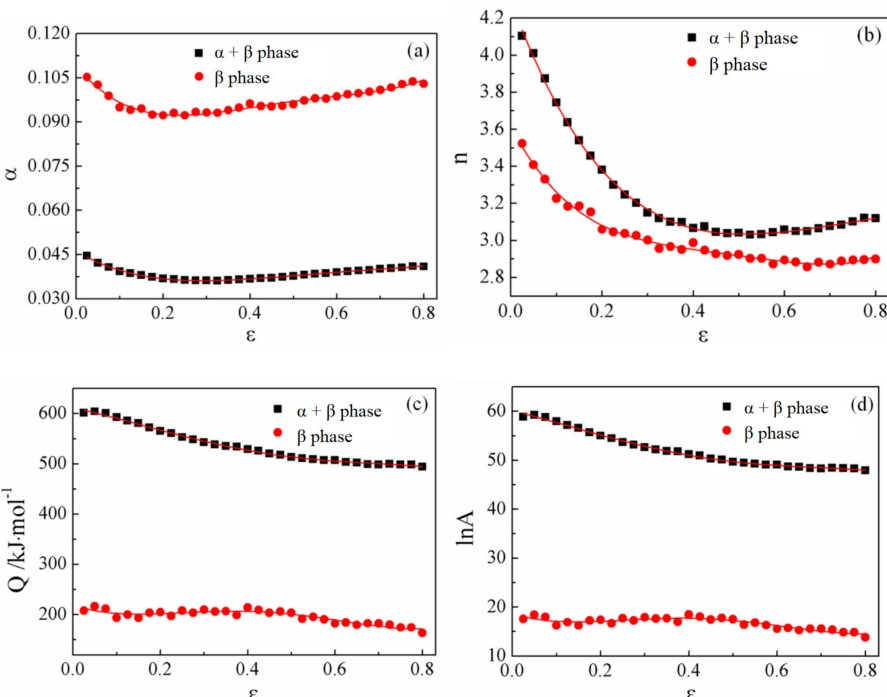

**Figure 13.** Relationships between (**a**) $\alpha$, (**b**) $n$, (**c**) $Q$, and (**d**) ln$A$ and true strain by polynomial fit of Ti-6.5Al-3.5Mo-1.5Zr-0.3Si titanium alloy.

By now, the polynomial coefficients of all material constants considering the effect of strain are calculated, and the flow stress at a certain strain can be estimated. Using the definition of the hyperbolic law, the constitutive equation that reorders Equation (5) can be expressed as follows:

$$\sigma = \frac{1}{\alpha} \ln \left\{ \left( \frac{\dot{\varepsilon}}{A} e^{Q/RT} \right)^{1/n} + \left[ \left( \frac{\dot{\varepsilon}}{A} e^{Q/RT} \right)^{2/n} + 1 \right]^{1/2} \right\}. \tag{14}$$

### 3.2.3. Verification of Constitutive Models

In this research, the aforesaid material constants of Ti-6.5Al-3.5Mo-1.5Zr-0.3Si titanium alloy are substituted into Equation (14), and then, the predicted flow stress curves can be obtained over the entire deformation temperature and strain rate. To verify the validity of the established modified constitutive model, the comparisons between the predicted and the experimental flow stress curves are carried out, as shown in Figure 14. The results indicate that the predicted flow stresses well agree with the experimental ones. Furthermore, the correlation coefficient (*R*), root mean square error (*RMSE*), and the average absolute relative error (*AARE*) are used to evaluate the accuracy of the established constitutive relationship model. Basically, the correlation coefficient is used to estimate the soundness of the linear relationship between the predicted and experimental values. Meanwhile, *RMSE* is the standard error and *AARE* can be considered to be an unbiased statistic for assessing the effectiveness of the established model. In general, the well performance of the established constitutive relation can be pledged when the values of *RMSE* and *AARE* are at a low level. They can be expressed as:

$$R = \frac{\sum_{i=1}^{N} \left( \sigma_r^i - \overline{\sigma}_r \right) \left( \sigma_t^i - \overline{\sigma}_t \right)}{\sqrt{\sum_{i=1}^{i=1} \left( \sigma_r^i - \overline{\sigma}_r \right)^2} \sqrt{\sum_{i=1}^{N} \left( \sigma_t^i - \overline{\sigma}_t \right)^2}} \tag{15}$$

$$RMSE = \sqrt{\frac{1}{N}\sum_{i=1}^{N}\left(\sigma_r^i - \sigma_t^i\right)^2} \tag{16}$$

$$AARE(\%) = \frac{1}{N}\sum_{i=1}^{N}\left|\frac{\sigma_r^i - \sigma_t^i}{\sigma_r^i}\right| \times 100 \tag{17}$$

where $\sigma_r$ is the predicted flow stress and $\sigma_t$ is the theoretical flow stress, and $\overline{\sigma}_r$ and $\overline{\sigma}_t$ are the mean values of $\sigma_r$ and $\sigma_t$, respectively. $N$ is the total number of data employed in this research. As shown in Figure 15, the values of correlation coefficient, $RMSE$, and $AARE$ were calculated to be 0.987, 3.585 MPa, and 9.62% in the β phase region and 0.979, 18.78 MPa, and 9.16% in the α + β phase region by regression analysis, respectively. It can also be found both outcomes suggest that the established modified-constitutive model of Ti-6.5Al-3.5Mo-1.5Zr-0.3Si titanium alloy is available to describe the flow stress behavior at elevated temperature and can be utilized for finite element numerical simulation of hot working for this alloy.

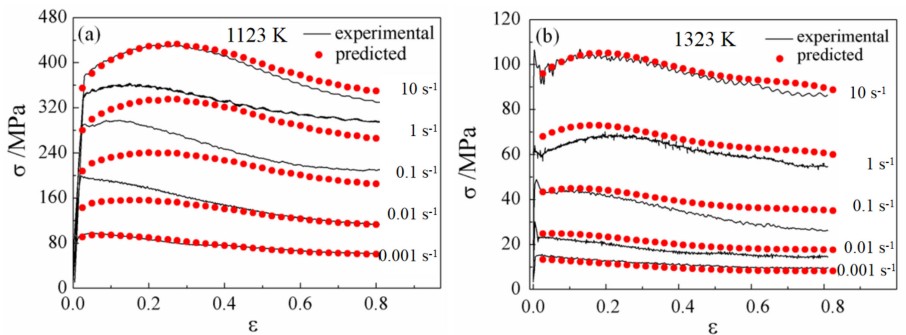

**Figure 14.** Comparison of the theoretical and the predicted values of Ti-6.5Al-3.5Mo-1.5Zr-0.3Si titanium alloy at (**a**) 1123 K and (**b**) 1323 K.

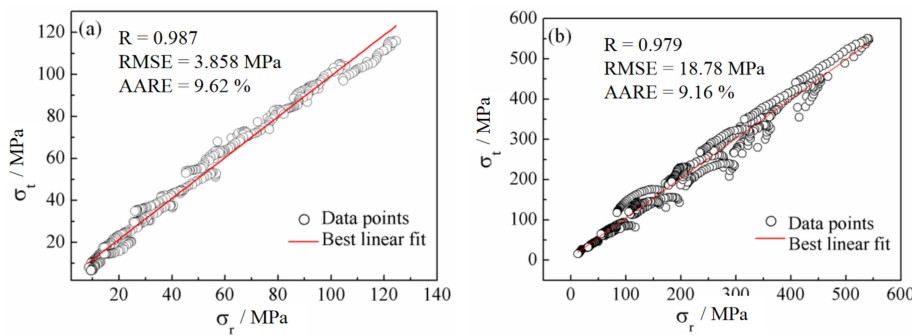

**Figure 15.** Error analysis between the theoretical and predicted values at (**a**) the α + β phase region and (**b**) the β. phase region.

## 4. Conclusions

In this work, Ti-6.5Al-3.5Mo-1.5Zr-0.3Si titanium alloy was isothermally compressed in the temperature ranging from 1073 to 1373 K at an interval of 50 K, and the strain rate range of 0.001–10 s$^{-1}$. Microstructure evolution and deformation behavior of this alloy are investigated in detail, and the following conclusions can be obtained.

1.　The flow stress is particularly sensitive to temperature and strain rate. The flow stress curves present a typical flow softening behavior in the α + β phase region, whereas such characteristic is weakened in the β phase region. The softening behavior can be explained by deformation heating effect and microstructure changes.

2. The deformation heating effect is caused by temperature rise during hot compression, and it is influenced by strain rate and deformation temperature. Temperature rise increases with the increasing strain rate and decreasing deformation temperature.

3. Microstructure is consisted of the primary α phase and transformed β matrix when materials are deformed below the β transus temperature. The fractions of the primary α phase decrease with the increase of deformation temperature and strain rate. The EBSD analysis shows that dynamic recovery may be the main mechanism for microstructure evolution below the β transus temperature. The fully phase transformation occurs for material deformed above the β transus temperature. The phase transformation is governed by Burgers orientation relations.

4. The material constants were obtained by the hyperbolic sine type constitutive equations, which have been expressed as functions of strain with fourth-order polynomials fit. In addition, the values of $n$, $Q$, and $\ln A$ were calculated in the β and α + β phase regions, respectively. The established constitutive model has a high prediction accuracy and is suitable for Ti-6.5Al-3.5Mo-1.5Zr-0.3Si titanium alloy.

**Author Contributions:** Writing—original draft, W.Y.; Data curation, W.J.; Writing—review and editing, W.J. and J.X.; Investigation, Z.Z.; Visualization, A.H.; Formal analysis, A.H., L.Q. and H.H.; Validation, J.X. All authors have read and agreed to the published version of the manuscript.

**Funding:** The authors are grateful for funding support from National Natural Science Foundation of China (NO. 51905436).

**Data Availability Statement:** All raw data supporting the conclusion of this paper are provided by the authors.

**Conflicts of Interest:** The authors declare no conflict of interest.

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
