# Peer review of "Analysis of Deformation Behavior and Microstructure Changes for α/β Titanium Alloy at Elevated Temperature"

_metals, doi:10.3390/met11020303_

Round 1
Reviewer 1 Report
The manuscript "Analysis of deformation behavior and microstructure changes for an α/β titanium alloy during elevated temperature" has been reviewed.
It deals with microstructure evolution and deformation behaviour of a Ti alloy. A constitutive model is proposed for the prediction of the flow stress behaviour.
The manuscript is quite clear. However in literature a lot of constitutive models can be found about Ti alloys. References must be integrated with more recent articles. In particular the advantages of the suggested model and the limitations should be highlighted at the end of the introduction, in discussion and in conclusions. This is the main lack of this manuscript.
At the same time some small adjustments are required, as detailed in the following:
Line 77: strain, not stain.
Line 149: temperature, not temperate.
Line 101: the observations were, not was...
Line 226: relationships, not relations.
Sentence line 120 - 123 is obscure, please explain better!
Increase readibility of Fig. 11, 14, 15.
Due to the high number of acronyms adopted in the manuscript, nomenclature at the beginning is warmly recommended.
Author Response
Response: Thank you very much for the valuable comments. In the revised manuscript, references have been updated. The features of the suggested model have been highlighted in “Introduction”, “Discussion” and “Conclusions”. We hope this manuscript is now up to the standard of “Metals”.
We have checked the English again and corrected some grammatical mistake and incorrect usages of words. Nomenclatures have been added at the beginning of this manuscript to explain acronyms. We hope this manuscript is now up to the standard of “Metals”.
Reviewer 2 Report
The manuscript is quite well prepared, however it contains a lot of linguistic and terminology errors. Please correct it considering following remarks:
Title – “ … during elevated temperature”? - it should be “… at elevated temperature”
Line 14 – Abstract – “Microstructure evolution and deformation behavior are investigated” – in my opinion “… were investigated”.
Line 79 – “ … the optical microscope (OM), …” – the electron microscope is optical too (but it has electron optics). You should use a term “light microscope (or microscopy)” and (LM) abbreviation – in whole text of your manuscript.
Lines 86 and 87 – “…a typical dual-modal microstructure …” – the term “bi-modal” is more suitable and generally accepted in the scientific literature.
Lines 97 and 99 – “The deformed specimens were sectioned parallel to the compression axis for microstructure analysis.” – Which area of mentioned cross-section was analysed by microscopic methods? It is especially crucial for compressed specimen. This should be mentioned in the text.
Line 99 – “The samples for optical metallographic examination …” – The samples for metallographic examination … “ is enough!
Lines 101 and 102 – “…by the OM, SEM and 101 EBSD.” – “…LM, ...” according my previous comment. Moreover, you should add some information about the equipment you used in described investigation (LM? SEM?).
Line 177 – “Table 2. Change of temperature rise of the material under different strain rate”??? – In my opinion, you present the temperature rise for different deformation temperature and strain rate. I suggest to write “Table 2. Deformation heating effect -the temperature rise (K) - of the material under different deformation conditions”. I suggest to supplement the table, as follows:
---------------------------------------------------------------------------------
Deformation temperature, K Strain rate, s-1
---------------------------------------------------------------------------------
and put values only (without units)
Line 178 – “3.1.3. Microstructure evolution” sounds much better!
Lines 205 and 206 – There is no “OM morphology”! Morphology relates to phase constituents! I suggest to write “The LM microstructure of ….”. Moreover, presented microstructure developed after deformation at: …” – not “during”! (especially in case of fig. 4d)
Lines 211 and 212 – as above – “The SEM microstructure of …. after deformation at: …”
Lines 213-228 – The term “distribution of misorientation” is unclear! You should specify it: angle misorientation or boundary misorientation?
Lines 233 and 234 – as before “Figure 6. … after deformation at: ……”
Line 238 – “Figure 7. The distribution of ANGLE misorientation …”?????
Lines 252 and 253 – Figure 8. – “… under different deformation conditions.” or “… after deformation under various conditions.”??? – The second variant is true, in my opinion.
Line 377 – “ … alloy WAS isothermally compressed … “ – not “is”.
Please pay more attention for tenses you use in the text of your manuscript. I suggest to check the text for language carefully before submitting the revised version.
Author Response
Response: Thank you for the referee’s good advice. We have checked the English again and corrected some grammatical mistake and incorrect usages of words. In addition, we have asked several colleagues who are skilled authors of English language paper to check the English. We hope this manuscript is now up to the standard of this journal.
Thanks again for reviewer’s kind advice for our manuscript. The manuscript has reached a new height in the light of the reviewer’s comments.
Reviewer 3 Report
In this manuscript, the authors investigated the microstructure evolution and deformation behavior of Ti-6.5Al-3.5Mo-1.5Zr-0.3Si titanium alloy. Moreover, a constitutive model has been developed. I think the results will be of interest to researchers in this area.
I have the following comments on the manuscript:
Experimental materials and procedure. The experimental procedure section is very poor. There is no information about optical and electron microscopes. There is also no information on the EBSD analysis technique (Scan step, analysis area).
Fig. 6. «…Fig. 6 illustrates the inverse pole figure (IPF)…». I think in this case must be « inverse pole figure (IPF) maps».
Fig. 6. «The IPF of Ti-6.5Al-3.5Mo-1.5Zr-0.3Si alloy during various deformation conditions». Why during? Maybe after various deformation regimes?
Figure 7a does not correlate with Figure 6a. The misorientation distribution shows only low-angle boundaries. Figure 6 shows the high-angle boundaries.
Fig. 6. Moreover, the high-angle and low-angle grain boundaries are not indicated in Figure 6 for all deformation regimes.
Fig. 7. What is the reason for such a strange distribution of grain misorientation angles? Why do distributions contain gaps? Maybe this is due to the small number of analyzed grains? It is necessary to clarify the Experimental procedure.
The authors say that the evolution of the microstructure during deformation has been investigated. Why is the microstructure before deformation not performed (only after heating)?
Line 363. «AARE were calculated to be 0.987, 3.585MPa and 9.62% in α phase region…» I think this is a mistake. It is necessary to specify the β phase region.
Author Response
Response: Thank you for the referee’s good advice. In the revised manuscript, the relevant experimental information have been added.
“Fig. 6 illustrates the inverse pole figure (IPF)…” has been revised as “Fig. 6 illustrates the inverse pole figure (IPF) maps…”.
The name of Fig. 6 has revised as “Fig.6. The inverse pole figure (IPF) maps of Ti-6.5Al-3.5Mo-1.5Zr-0.3Si alloy after deformation at: (a) 1123K /0.1s-1, (b) 1223K /0.1s-1 and (c) 1323K /0.1s-1.”
There are the high-angle boundaries in Fig. 7a, for example, at the locations of 30°, 40° and 50°. However, they are a very small percentage and not easy to observe.
In the revised manuscript, all the high-angle and low-angle grain boundaries are indicated in Figure 6 for all deformation regimes. The bold black lines correspond to high-angle boundaries with misorientation over 15° while the thin black lines represent low-angle boundaries with misorientation between 2° and 15°.
This distribution of grain misorientation angles is not strange, At higher temperature, the appearance of the boundary misorientation nearby 60° and 90° is associated with the precipitation of near alpha phase, which is governed by Burgers orientation relationships. The gaps in Fig. 6 is beta phase and unrecognized area, which are not the main research content in this manuscript. The experimental procedure has been clarified in the revised manuscript.
Microstructure after heating without deformation is not performed in this manuscript. This work only involves the effect of deformation on microstructure. Before deformation, the specimens are heated to deformation temperature and kept for only 3min, which minimizes the effect of heating on microstructure.
“…were calculated to be 0.987, 3.585MPa and 9.62% in α phase region…” is a mistake. It has been revised as “…were calculated to be 0.987, 3.585MPa and 9.62% in β phase region…”.
Round 2
Reviewer 1 Report
Accept as it is
Reviewer 3 Report
Many thanks to the authors for revising the manuscript. I think that the manuscript can be accepted for publication.